# Wireless LC Conformal Temperature Sensor Based on Ag Film (9912-K FL) for Bearing Temperature Measurement

**DOI:** 10.3390/nano12172899

**Published:** 2022-08-23

**Authors:** Chen Li, Qiyun Feng, Yingping Hong, Lixia Gao, Ximing Guo, Wenzhi Xue, Jijun Xiong

**Affiliations:** 1State Key Laboratory of Dynamic Measurement Technology, North University of China, Taiyuan 030051, China; 2Science and Technology on Electronic Test and Measurement Laboratory, North University of China, Taiyuan 030051, China; 3Sichuan Gas Turbine Research Establishment of AECC, Chengdu 610500, China

**Keywords:** Ag film (9912-K FL), wireless sensor, bearing temperature, screen-printed, temperature sensor

## Abstract

As the key component of aero-engines and industrial gas turbines, a bearing’s working temperature at high speed is close to 300 ℃. The measurement of an engine bearing’s temperature is of great significance to ensure flight safety. In this study, we present a wireless LC conformal temperature sensor for bearing temperatures, which integrates silver on the bearing surface in situ through a screen-printing process. This process makes Ag film (9912-K FL) firmly adhere to the bearing surface and realizes wireless measurements for bearing temperatures in situ. A high-temperature holding experiment of the prepared sensor was conducted, and the results showed that the sensor can work stably for 10 h at 300 ℃. We tested the designed wireless LC conformal temperature sensor at 20–270 ℃. The results showed that the proposed temperature sensor attained as good accuracy and stability in the temperature range 20–270 ℃. The sensitivity of the temperature measurements was 20.81 KHz/℃  when the bearing rotateds, the maximum repeatability was 0.039%, the maximum uncertainty was 0.081%, and the relative error was stable within  0.08%.

## 1. Introduction

The temperature change of a bearing directly affects its lifespan and stability; thus, bearing condition monitoring technology is of great significance to the fault prediction and normal operation of engines [1,2,3,4,5,6]. Bearings are an important rotating component of engines. An engine bearing must bear a temperature of 150 ℃. In the future, engine bearings are expected to run stably above  250 ℃ to improve flight efficiency [7,8,9,10]. To prevent disasters caused by structural failure due to high temperatures, sensors must measure high temperatures in harsh environments; for example, for heat resistance monitoring of spacecrafts at high temperatures, high-temperature testing of rotary bearings, and rotation testing of high-speed shafts of aircraft engines, disc brakes, and jet engines [11,12,13]. The integration of wireless passive sensors on dynamic balance bearings for telemetry is a current development trend of bearing parameter measurement [14,15,16], which is integrated on a dynamically balanced ball bearing and operates in a test setup, simulating the operating conditions of a dual compressor turbocharger [17]. Wireless micro electromechanical systems (MEMS) temperature sensors, with a high working temperature, constant sensitivity, and high manufacturing yield, are widely used in measurement systems [18,19,20]. Scott et al. [21] introduced a wireless MEMS dual piezoelectric wafer temperature sensor that can dynamically test the actual bearing temperature value of helicopter bearings at different speeds. 

Wang [4] reported a fiber Bragg grating (FBG)-based sensor for the simultaneous measurement of a train bearing’s temperature. The sensor had a high temperature sensitivity of 35.165 pm/℃ at a temperature range 20–100 ℃. Henao-Sepulveda [5] presented a wireless technique to directly monitor the temperature of bearing cages at a temperature range 20–80 ℃. The wireless transmission of signals was achieved using RF technology. Draney [8] used a miniaturized prototype and an oscillator/modulation-based sensing approach. By making the oscillator components temperature-dependent, temperature data were frequency modulated onto the oscillator signal. Janssens [22] proposed a novel automatic fault detection system using infrared imaging, focusing on bearings of rotating machinery. The system was able to distinguish between eight different conditions with an accuracy of 88.25%. Joshi [23] proposed two temperature telemeters as a condition-monitoring instrument for bearings. They can reflect the change in bearing cage temperature with rotational speed and torque, and the maximum temperature can reach 100 ℃. Yan [24] proposed a wireless temperature sensor based on printed circuit board (PCB) materials. The temperature-sensitive structure of the sensor comprised a dielectric-loaded resonant cavity and a patch antenna for the transmission of temperature signals. Zhou [25] proposed a measurement method for the temperature distribution of the inner and outer rings of a bearing based on the fiber Bragg grating, and a bearing test rig was established to measure the temperature of the bearing rings at multiple points.

The commonly used wireless temperature sensing systems (temperature sensors) include surface acoustic waves (SAW), microwave wireless temperature sensors, and inductance and capacitance (LC) [26,27,28,29,30]. The temperature sensor based on the resonant inductance capacitance (LC) circuit did not require a power supply or physical lead connection [31,32]. Because of the small size and stability of LC sensors, they are especially suitable for short-distance transmission tests in harsh industrial environments. Thus far, several studies have successfully promoted wireless LC-based sensing technology and expanded its applications in many fields [33,34,35,36]. MD [37,38] introduced a new type of high temperature dynamic viscosity sensor that is used for on-site condition monitoring of engine lubricating oil. The sensor is also applicable for RF ball bearing cage temperature telemeters in the high-speed turbocharger, in which the temperature-dependent LC energy storage circuit is attached to the bearing cage, and the fixed receiving coil of the axial displacement is used for wireless monitoring.

In this study, a wireless LC conformal temperature measurement sensor was demonstrated for the temperature measurement of bearings. Through a screen-printing process, the Ag film was firmly integrated on the bearing surface in situ. The stable electrical properties of the Ag film at high temperatures allowed the sensor to maintain a high accuracy and stability in the test temperature range of 20–270 ℃. The high temperature rotating experimental platform also verified the superior performance of the designed wireless LC conformal temperature sensor.

## 2. Experiment

### 2.1. Materials

ESL 9912-K FL is a fine-line printing silver conductor with a wide range of applications, such as chip resistors, consumer hybrids, potentiometers, and heaters. It exhibits excellent line resolution printing with 75 micrometer wide lines. Due to the wide firing temperature range, this conductor may be processed onto a variety of substrates including glass, porcelain enameled steel (PES), alumina, and special ceramics. The levelling time of ESL 9912-K FL is 5–10 min at 20 ℃ and the drying time is 10–15 min at 125 ℃. Furthermore, the firing temperature is 850 ℃ on alumina/beryllia/ceramics.

### 2.2. In Situ Fabrication of a Wireless LC Conformal Temperature Sensor

As shown in Figure 1, preparation of the wireless LC conformal temperature sensor included the following steps: 

Design the pattern and size of the metal screen for printing, with 325  metal screen meshes, an aperture size of  45  microns, and a thickness of 20 microns;

A.Clean the test bearing surface with alcohol and a dust-free cloth. Integrate silver paste (9912-K FL) in situ on the surface of the inner ring of the test bearing using the screen-printing process to form a spiral inductor with an inner diameter of 20 mm, outer diameter of 24 mm, line width of 1 mm, and a line pitch of 1 mm, with two turns in total;B.Place the test bearing into a muffle furnace for sintering and maintain 850 ℃ for 30 min;C.After completion, take out the bearing and let it stand for cooling, and use a multimeter to measure whether the coil is conducting;D.Connect the test bearing with the ceramic rod using a high-temperature-resistant non-conductive adhesive and place the connected test bearing and ceramic rod on the heating table at  80 ℃  for  2 h followed by  150 ℃  for  3 h. The ceramic rod is 20 mm in diameter and 350 mm in length;E.Construct the antenna using 1 mm diameter copper wire; the antenna diameter is approximately  40 mm, and the antenna utilizes an SMA-KE deflection angle connector;F.Connect one end of the ceramic rod to the coupling of the motor and the other end to the supporting bearing.

**Figure 1 nanomaterials-12-02899-f001:**
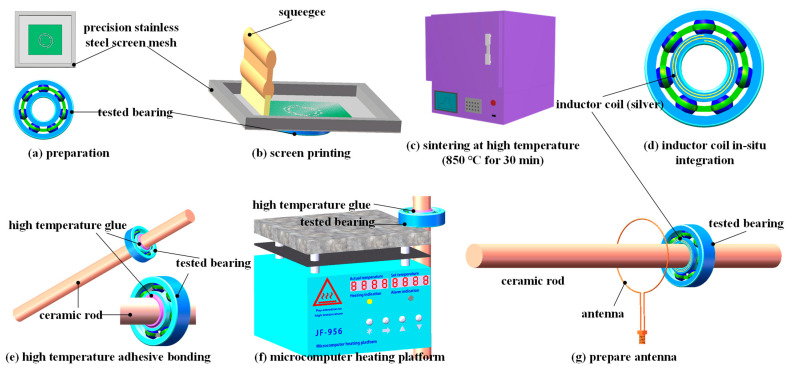
Fabrication of a printed temperature sensor using screen printing.

## 3. Results and Discussion

The test bearing after the sintering of the Ag film is shown in Figure 2a. The film thickness was 13.64 μm, as shown in Figure 2b. The Ag film was tightly connected to the bearing base, as shown in Figure 2c. To study the microstructure of the prepared LC temperature sensor, the characteristics of Ag Films under different high temperature holding conditions were observed. The surface morphology and microstructure of Ag films after high temperature sintering were observed using scanning electron microscopy (SEM), as shown in Figure 3. Figure 3a shows the SEM images of the Ag film at different magnifications. The Ag film prepared by screen printing was formed by bonding ultrafine silver powder with an organic binder. In the sintering process, the organic components in the film gradually volatilized, and the ultrafine silver powder combined at high temperature to form a dense network structure. Figure 3b shows the SEM image of the Ag film after long-term high-temperature testing. With the increase in test duration, the gap between the Ag grains decreased and the grains fused better, forming a denser Ag film. The crystal structure is shown in the X-ray diffraction (XRD) diagram in Figure 4. Through the analysis of the mode using the Origin software, the diffraction peak was confirmed at 2θ. The purity of the silver formation at 38.2°, 44.4°, 64.6°, and 77.59° corresponded to the (111), (200), (220), and (311) planes of the face-centered cubic (FCC) structure, respectively. In addition, it was confirmed that at 2θ, the purity of alumina formed at 25.62°, 35.21°, 52.64°, 57.60°, 61.41°, and 68.34° corresponded to the (110), (112), (220), (132), (332), and (130) planes of the face-centered cubic (FCC) structure, respectively. Moreover, comparing the XRD patterns at 3 h and 10 h of sintering, the purity levels of silver and alumina were essentially unchanged. This demonstrates that the metallic silver can function stably at a high temperature for an extended duration [39]. In addition, the sensor was characterized using an X-ray diffractometer (Japan Rigaku Miniflex), in which the scanning angle ranged from 20° to 80°, the scanning speed was 2 °/min, and the test target was copper.

To verify the performance of the wireless LC conformal temperature sensor, we constructed a platform for the wireless measurement of bearing temperatures based on an LC temperature sensor, which mainly comprised a motor, antenna, test bearing, support bearing, thermocouple, heating plate, temperature control panel, and network analyzer, as shown in Figure 5a. The antenna, test bearing, and supporting bearing were placed in a muffle furnace, and the motor was kept at room temperature. The muffle furnace controlled the ambient temperature of the antenna and test bearing, and the network analyzer (starting frequency of 135 MHz, cut-off frequency of 160 MHz, and 2000  measurement points) was used to measure the coupling between the antenna and test bearing at different temperatures and rotating speeds. The signal was transmitted wirelessly by testing the electromagnetic coupling between the antenna and LC circuit. As the temperature varied, the capacitance changed, which in turn led to changes in the characteristic frequency of the LC equivalent circuit [40,41,42]. The antenna and bearing transmitted signals wirelessly through the electromagnetic coupling between coils, as shown in Figure 5b. The coupling between the antenna and coil at different angles was simulated using the ANSYS software, as shown in Figure 5c. The simulation results showed that coupling at different angles had almost no influence on the characteristic frequency of the antenna, which indicated that the designed wireless LC conformal temperature sensor can realize wireless temperature measurement in a rotating environment. When the sensor was coupled with the antenna, the reflection coefficient (Γ) at the antenna end, S_11_ parameter, impedance ($Z_{in}$) and sensor resonance frequency (f) can be expressed as Equations (1)–(4) [43,44]. The circuit equivalent model of the sensor is shown in Figure 5b, where $R_{a}$, $L_{a}$, and $C_{a}$ represent the equivalent resistance, equivalent inductance, and equivalent capacitance of the antenna equivalent circuit, respectively. $R_{s}\;$, $L_{s}\;$, and $C_{s}$ represent the equivalent resistance, equivalent inductance, and equivalent capacitance of the LC equivalent circuit, respectively. M represents the mutual inductance coupling coefficient between the two circuits.
(1)$$\Gamma ={\frac{Z_{in}-Z_{0}}{Z_{in}+Z_{0}}|}_{Z_{0}=50\Omega }$$
(2)$$S_{11}=20\log(\left|\Gamma \right|)$$
(3)$$Z_{in}=R_{a}+j\omega L_{a}+\frac{{\left(\omega M\right)}^{2}}{R_{s}+j\omega L_{s}-\frac{j}{\omega C_{s}}}$$
(4)$$f=\frac{1}{\sqrt{L_{s}C_{s}}}$$

According to Equations (1)–(4), a change of ambient temperature in the sensor will lead to a change of the characteristic frequency and S_11_. Thus, by analyzing the frequency and S_11_ parameter, the temperature signal in the high temperature rotational environment can be wirelessly measured. The S_11_–frequency parameter curve of the antenna terminal at different rotating speeds at room temperature is shown in Figure 6a. By observing and analyzing the graph, it can be seen that the frequency and S_11_ parameter fluctuated slightly with the rotating speed, which proves that the proposed wireless LC conformal temperature sensor is effective in the wireless temperature measurement of bearings. We measured the temperature at 20–270 ℃ at 0–60 rpm on the developed test platform and obtained the S_11_–frequency parameter curves of the antenna at different temperatures using the network analyzer, as shown in Figure 6b. By observing the change law of the curve, it can be seen that the characteristic frequency of the antenna decreased gradually and the S_11_ parameter increased gradually with an increase in temperature. 

The frequency variation with temperature at different rotating speeds was measured. The fitting curve of discrete points of the frequency variation with temperature at different rotating speeds is shown in Figure 7a. Variation of the S_11_ parameter with temperature at different rotating speeds was also measured. The fitting curve of discrete points of the S_11_ parameter with temperature at different rotating speeds is shown in Figure 7b. It can be seen from the graph that the frequency variation with temperature demonstrated good linearity at different rotational speeds, while S_11_ had poor linearity with temperature. In addition, the sensitivity of the frequency variation with temperature was  k=20.68 kHz/℃, and the sensitivity of the S_11_ parameter variation with temperature was k=0.0067 dB/℃.

The frequency at different rotational speeds and the stability of S_11_ with temperature were analyzed, as shown in Figure 8. From the observation curves, it can be seen that frequency and S_11_ were stable with a change in temperature in the static state (0 rpm), and the stability of frequency was better than that of S_11_. In a rotating environment (60 rpm), the frequency maintained good stability with a change in temperature, while S_11_ showed significant fluctuations with a change in temperature. Therefore, in this study, we characterized the temperature by analyzing the frequency change.

The nonlinear error ($e_{n}$) of the sensor was analyzed, and five groups of tests were conducted at different rotational speeds within 20–270 °C, as shown in Figure 8. Linearity is an important index used to describe the static characteristics of a sensor, and the nonlinear error ($e_{n}$) was calculated based on a fitted straight line. By considering the linear fitting of data as a reference line, the nonlinear error of the sensor at different rotational speeds was analyzed. As can be seen from Figure 9, the frequency maintained good linearity with a change in temperature at different rotational speeds, and the sensitivity of the sensor increased slightly with an increase in rotational speed.

Subsequently, the repeatability ($e_{R}$) and uncertainty (U) of the sensor were analyzed, and five groups of tests were conducted at different rotational speeds within 20–270 °C, as shown in Figure 10 and Figure 11. The repeatability and uncertainty represented the dispersion and randomness of the sensor test results, which were important indices to determine whether the sensor operated stably over time. The sensor’s repeatability error ($e_{R}$) was expressed as a percentage comparing the maximum non-repetition output error (Δmax) and full-scale output ($y_{ES}$). Uncertainty (U) included Class A uncertainty ($U_{A}$) and Class B uncertainty ($U_{B}$). $U_{A}$ was caused by a measurement error, whereas $U_{B}$ was caused by an instrument error. The maximum repeatability ($e_{R}$) was  0.039 %, and the maximum uncertainty (U) was 0.081 % (170 ℃, 0 rpm). The repeatability ($e_{R}$) and uncertainty (U) values of several groups of temperature tests at different rotational speeds did not fluctuate noticeably with an increase in rotational speed, which indicates that the rotational speed caused little interference to the wireless temperature measurement and proved the effectiveness of the proposed wireless LC conformal temperature sensor in a rotating environment. At the same speed, repeatability ($e_{R}$) and uncertainty (U) did not change noticeably with an increase in temperature, which indicates that the proposed wireless LC conformal temperature sensor maintained good stability throughout the temperature range in a rotating environment. 

Further, the relative error of frequency variation with temperature at different rotational speeds was analyzed, as shown in Figure 12. The relative error of frequency was stable within  0.08 % for increasing temperature and rotational speed. The proposed wireless LC conformal temperature sensor had a high measurement accuracy throughout the temperature range and was not affected by rotational speed.

In order to further prove the stability of the designed sensor in long-term high-temperature measurement, we carried out a high-temperature holding experiment at 300 °C for 10 hours and recorded the frequency characteristic curve of the antenna every hour with a network analyzer. As shown in Figure 13, the frequency still kept good stability under long-term high-temperature measurements, while S_11_ gradually increased with time.

## 4. Conclusions

In this study, we proposed a wireless measurement method for a conformal temperature sensor for bearing. The Ag film was firmly integrated on the bearing surface in situ using a screen printing process. The temperature signal was transmitted wirelessly through electromagnetic coupling between the antenna and inductance coil. A change in the test bearing’s temperature led to a change in the characteristic frequency of the antenna, which was measured using a network analyzer. The high-temperature holding experiment was conducted to verify whether the sensor could function stably in high-temperature environments. The results showed that the proposed wireless LC conformal temperature sensor could function stably at 300 ℃ for 10 h. The LC temperature sensor was tested at 20–270 ℃ with a high temperature rotating experimental platform. The results showed that the sensor achieved good linearity and accuracy in under bearing rotation. The sensitivity of the sensor was 20.81 KHz/℃, and the relative error was kept within 0.08 %. The sensor could function stably during the test, with a maximum repeatability of 0.039 % and maximum uncertainty of  0.081%. Compared with previous wireless bearing temperature measurement systems, the wireless LC conformal temperature sensor proposed in this study had a higher temperature measurement range and accuracy and successfully functioned under the measured conditions for an extended period.

## Figures and Tables

**Figure 2 nanomaterials-12-02899-f002:**
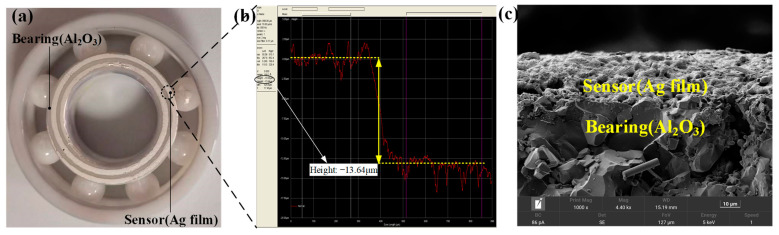
(**a**) Physical bearing after preparation; (**b**) measured thickness of the Ag film using a step meter; (**c**) SEM diagram of the Ag film and bearing cross section.

**Figure 3 nanomaterials-12-02899-f003:**
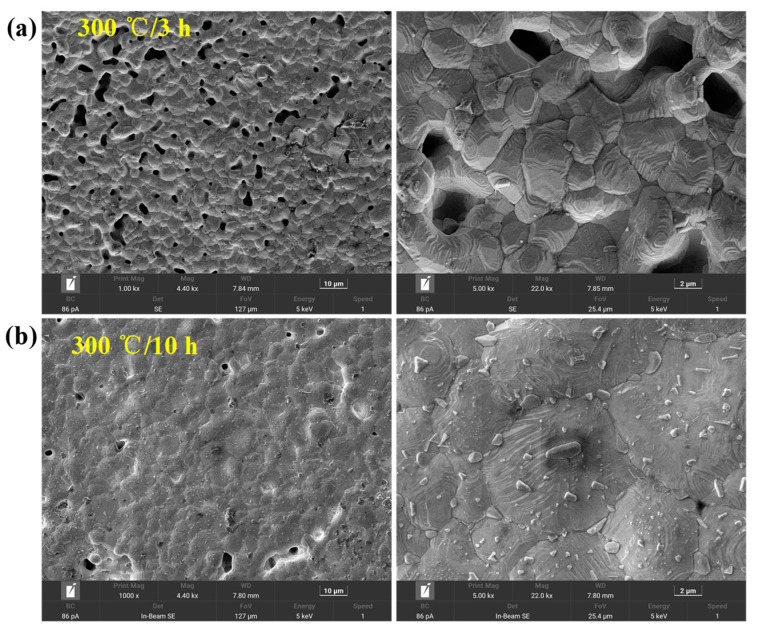
(**a**) SEM of a wireless LC conformal temperature sensor at 300 °C for 3 h; (**b**) SEM of a wireless LC conformal temperature sensor at 300 °C for 10 h.

**Figure 4 nanomaterials-12-02899-f004:**
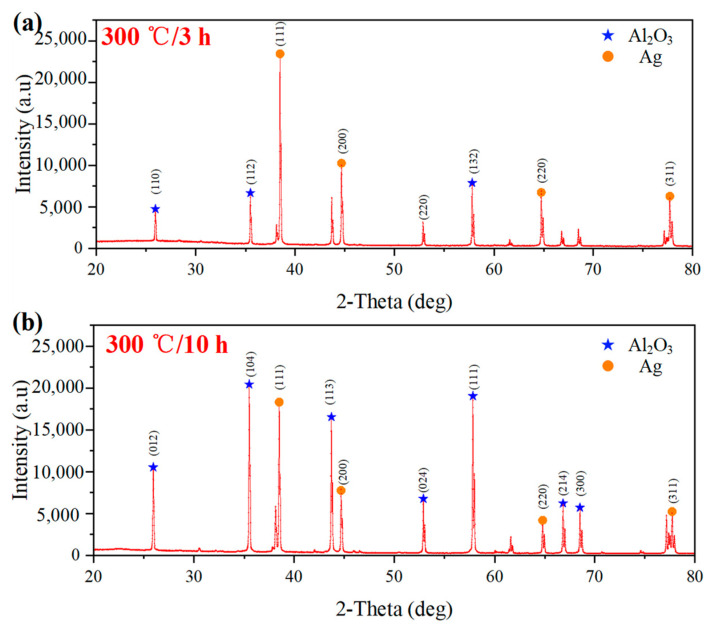
(**a**) XRD of a wireless LC conformal temperature sensor at 300 °C for 3 h; (**b**) XRD of a wireless LC conformal temperature sensor at 300 °C for 10 h.

**Figure 5 nanomaterials-12-02899-f005:**
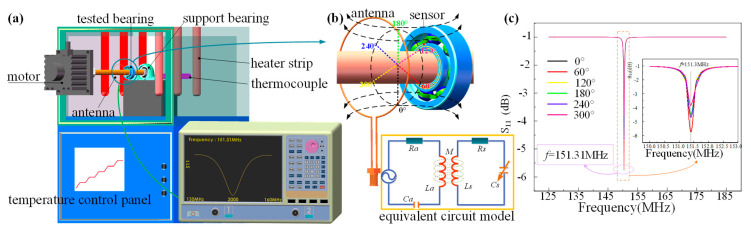
(**a**) Wireless LC temperature sensor and its application in a wireless measurement system of bearing temperature; (**b**) wireless transmission of temperature signal; (**c**) coupling between the antenna and coil at different angles simulated using ANSYS.

**Figure 6 nanomaterials-12-02899-f006:**
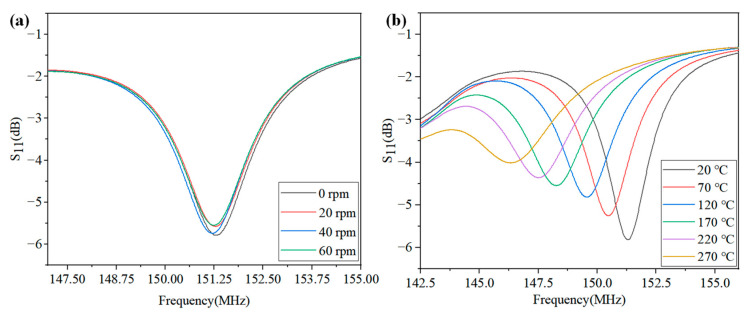
(**a**) S_11_-frequency characteristic curve at different speeds; (**b**) S_11_-frequency characteristic curve at different temperatures.

**Figure 7 nanomaterials-12-02899-f007:**
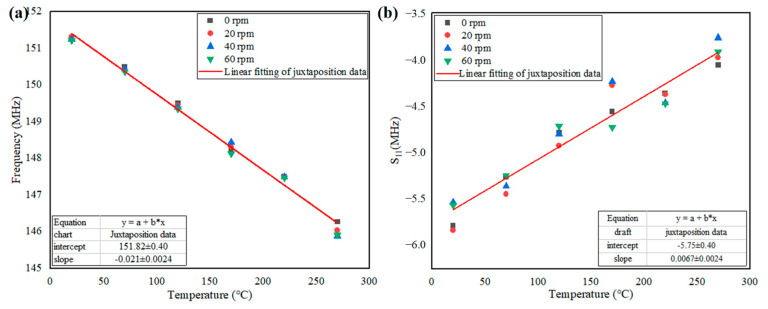
(**a**) Frequency at different speeds versus temperature; (**b**) frequency variation with temperature at different speeds.

**Figure 8 nanomaterials-12-02899-f008:**
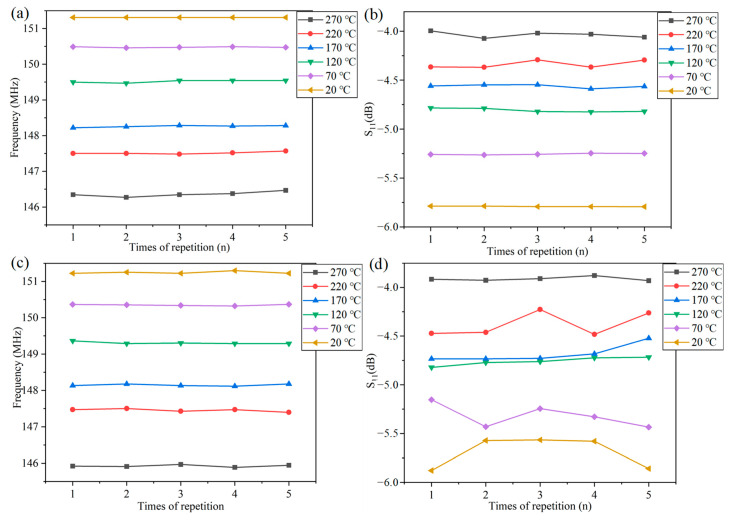
(**a**) Stability of frequency with temperature at  0  rpm; (**b**) stability of S_11_ with temperature at  0  rpm; (**c**) stability of frequency with temperature at 60 rpm; (**d**) stability of S_11_ with temperature at 60 rpm.

**Figure 9 nanomaterials-12-02899-f009:**
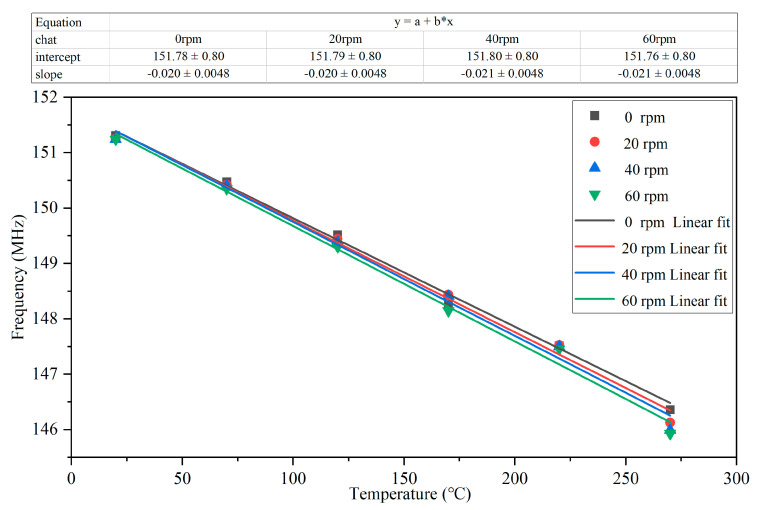
Comparison of the nonlinear error ($e_{n}$) values for five sets of speed measurements at different rotational speeds.

**Figure 10 nanomaterials-12-02899-f010:**
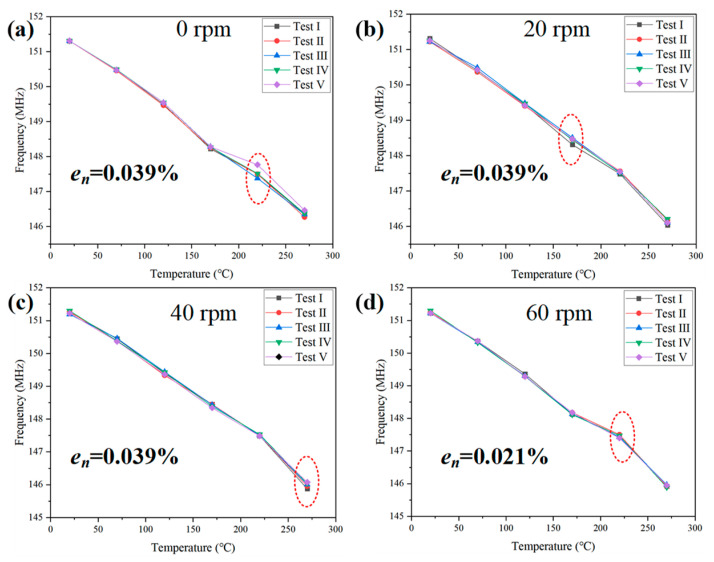
Repeatability of the wireless LC conformal temperature sensor at different rotational speeds. (**a**) 0 rpm; (**b**) 20 rpm; (**c**) 40 rpm; (**d**) 60 rpm.

**Figure 11 nanomaterials-12-02899-f011:**
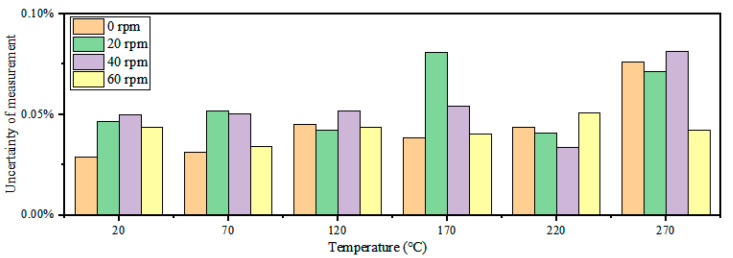
Uncertainty of the wireless LC conformal temperature sensor at different rotational speeds.

**Figure 12 nanomaterials-12-02899-f012:**
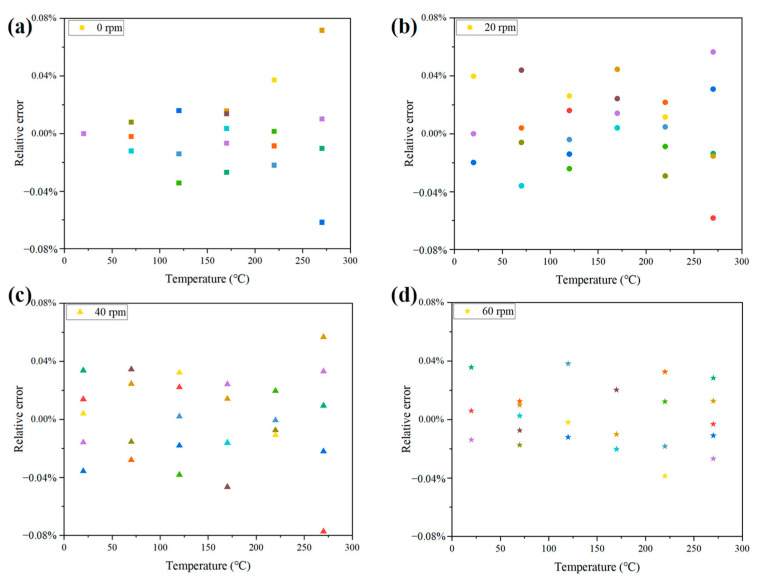
Relative error of the wireless LC conformal temperature sensor at different rotational speeds. (**a**) 0 rpm; (**b**) 20 rpm; (**c**) 40 rpm; (**d**) 60 rpm.

**Figure 13 nanomaterials-12-02899-f013:**
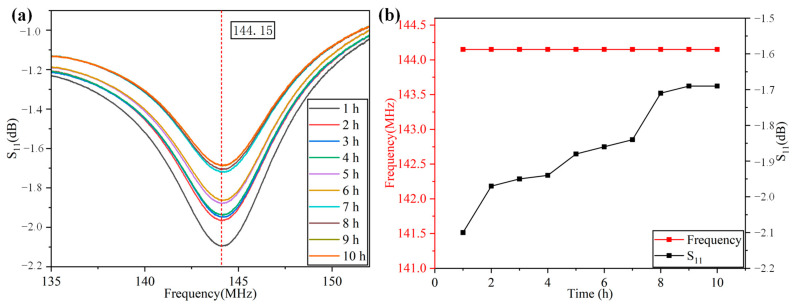
(**a**) S_11_-frequency characteristic curve in a high-temperature holding experiment at 300 °C for 10 hours; (**b**) frequency and S_11_ change with the length of time maintained at 300 °C.

## Data Availability

Not applicable.

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
