# Peer review of "Wireless LC Conformal Temperature Sensor Based on Ag Film (9912-K FL) for Bearing Temperature Measurement"

_nanomaterials, 2022, doi:10.3390/nano12172899_

Round 1

Reviewer 1 Report

This paper present a temperature sensor installed on a bearing.  It is an interesting topic, however, this type of studies have been conducted many times over the years and the authors neglect to include these references in their work.

The authors also neglect to show their actual bearing with details of sensor installed.  All figures except one are CAD drawing and anything can be shown on CAD, however, construction of such sensors are quite difficult.  

Due to these reasons, this reviewer has the following mandatory changes before submission for review.

For full details, please see the attached file containing suggestions and comments for the authors.

Author Response

Original Manuscript ID: nanomaterials-1836591

Original Article Title: LC conformal passive temperature sensor based on Ag film (9912-K FL) for bearing temperature measurement

TO: nanomaterials reviewer

RE: Response to reviewers

Dear reviewer,

Thank you for allowing a resubmission of our manuscript, with an opportunity address the reviewers’ comments.

We are uploading (a) our point-by-point response to the comments (below) (Responses to reviewer’s Comments), (b) an updated manuscript with yellow highlighting indicating changes, and (c) a clean updated manuscript without highlights.

Best regards,

<Chen Li> et al.

Responses to reviewer’s Comments

The authors would like to thank the reviewer for his/her help and efforts by providing constructive comments and valuable suggestions. Changes have been made in the revised manuscript to incorporate most of the reviewer’s suggestions. The changes and justifications for clarifying the reviewer’s comments are summarized as follows:

Reviewer #1:

Concern #1: This paper present a temperature sensor installed on a bearing. It is an interesting topic, however, this type of studies have been conducted many times over the years and the authors neglect to include these references in their work. Include all references shown below.

  • Joshi A, Marble S, Sadeghi F. Bearing cage temperature measurement using radio

telemetry[J]. Proceedings of the Institution of 308 Mechanical Engineers, Part J: Journal of Engineering Tribology, 2001, 215(5): 471-481.

  • A. Ashtekar, A. Kovacs, F. Sadeghi and D. Peroulis, "Bearing cage telemeter for the detection of shaft imbalance in rotating systems," 2010 IEEE Radio and Wireless

Symposium (RWS), 2010, pp. 5-8, doi: 10.1109/RWS.2010.5434255.

  • Cao, L, Sadeghi, F, Stacke, LE. A wireless sensor telemeter for in situ cage vibration measurement and corroboration with analytical results. Tribol Trans 2018; 61: 1013–1026.
  • M.D. Brouwer, L.A. Gupta, F. Sadeghi, D. Peroulis, D. Adams High temperature dynamic viscosity sensor for engine oil applications Sensors Actuators A Phys, 173 (1) (2012), pp. 102-107
  • Wireless Temperature Microsensors Integrated on Bearings for Health Monitoring Applications - Scott, S. et al. MEMS 2011
  • Early - Warning Wireless Telemeter for Harsh-Environment Bearings - Kovacs, A. et al. IEEE Sensors 2007
  • An Inherently Robust 300 C MEMS Temperature Sensor for Wireless Health Monitoring of Ball and Rolling Element Bearings - Scott, S. et al. IEEE Sensors 2009
  • High Reliable MEMS Temperature Sensors for 275 C Applications Part 1: Design and Technology - Scott, S. et al. Journal of Microelectromechanical Systems 2013
  • High Reliable MEMS Temperature Sensors for 275 C Applications Part 2: Creep and Cycling Performance - Scott, S. et al. Journal of Microelectromechanical Systems 2013
  • Application of Ball Bearing Cage RF Temperature Sensor in High Speed Turbocharger Brouwer, M. et al. IEEE 2014
  • Remotely Powered Wireless Strain Telemeter - Gupta, L. et al. IEEE 2014

Author response: Thank you very much for your constructive comments and valuable suggestions. At the suggestion of reviewers, we corrected the introduction and reference of this paper, according to the reference given by reviewers.

Author action: We updated the manuscript by adding  The integration of wireless passive sensors on dynamic balance bearings for telemetry is a current development trend of bearing parameter measurement [1416], which is integrated on a dynamically balanced ball bearing and operates in a test setup, simulating the operating conditions of a dual compressor turbocharger [17]. Wireless micro electromechanical systems (MEMS) temperature sensors, with a high working temperature, constant sensitivity, and high manufacturing yield, are widely used in measurement systems [1820]. Scott et al. [21] introduced a wireless MEMS dual piezoelectric wafer temperature sensor that can dynamically test the actual bearing temperature value of helicopter bearings at different speeds.  in the introduction.

We updated the manuscript by adding  Brouwer [37,38] introduced a new type of high temperature dynamic viscosity sensor that is used for on-site condition monitoring of engine lubricating oil. The sensor is also applicable for RF ball bearing cage temperature telemeters in the high-speed turbocharger, in which the temperature dependent LC energy storage circuit is attached to the bearing cage, and the fixed receiving coil of the axial displacement is used for wireless monitoring.  in the introduction.

Concern #2: The authors also neglect to show their actual bearing with details of sensor installed. All figures except one are CAD drawing and anything can be shown on CAD, however, construction of such sensors are quite difficult. Show pictures of constructed sensors on the bearing.

Author response: Fig. 2 (a) is a physical drawing of the sensor on the bearing. At the suggestion of the reviewer, we realized that the description of Figure 2(a) might not be clear enough, so we modified Figure 2(a).

Author action:

Figure 2. (a) Physical bearing after preparation; (b) Measured thickness of Ag film using a step meter; (c) SEM diagram of Ag film and bearing cross section.

We appreciate for reviewer’s warm work earnestly, and hope that the correction will meet with approval. Thanks for your constructive comment and valuable suggestions once again, and the authors hope to be able to learn more knowledge from you.

Reviewer 2 Report

The article is interesting, the work is accurate and clearly described, only a few typing errors and expressions have to be corrected; see below. Some points should be clarified to make the whole work more understandable for the reader.

1. Introduction reports: “… focussing on bearings of rotating machinery. It is able to distinguish between all eight different”, this expression should be corrected.

2. Introduction reports: “… So far, a large number of researches had successfully promoted LC-based passive wireless sensing technology and expanded its applications in many fields …”, this expression should be corrected.

3. Experiments 2.2 reports “…on the surface of the inner ring of the tested bearing by screen printing process to form a spiral inductor with an inner diameter of 20 mm and …”, this expression should be corrected.

4. The temperature unit “°C” in some case is close to the corresponding number, in other cases there is a space in between the number and the T unit, it should be uniform through the text.

5. Result and Discussion reports “… The Ag film is tightly connected with the bearing base, as shown in Figure 2 (c) …”, this expression should be corrected.

6. Conclusions reports “ … Compared with the previous wireless bearing temperature measurement system, the LC conformal passive temperature sensor proposed in this study has higher temperature measurement range and accuracy, and can work under the measured condition for a long time.” This expression should be corrected.

7. The authors affirm: “... the results show that the sensor can work steadily for 10 hours at 300 °C. …” This statement must be supported by the frequency response data, not only on the results obtained with x-rays and SEM.

8. More information on the Ag film (9912-K FL) should be included in the paper (Materials). The final film seems to be composed by Ag and Al2O3. Author may explain which is the role of Al2O3?

9. Table 1, reporting PARAMETERS OF XRD, is not essential as table, data can be included in the main text.

10. Section “Result and discussion” reports the reflection coefficient at the antenna end, parameter, impedance and sensor resonance frequency as 1-4 equations. All the symbols used in the equation must be explained.

11. By keeping the color, of figure 8, matched to the T specification, data comparison is more straightforward and less complicated.

12. Authors should clarify in the text how the nonlinear error of the sensor has been evaluated, since the fitting equations reported in figures 7 and 9 are the same, and its meaning.

Author Response

Original Manuscript ID: nanomaterials-1836591

Original Article Title: LC conformal passive temperature sensor based on Ag film (9912-K FL) for bearing temperature measurement

TO: nanomaterials reviewer

RE: Response to reviewers

Dear reviewer,

Thank you for allowing a resubmission of our manuscript, with an opportunity address the reviewers’ comments.

We are uploading (a) our point-by-point response to the comments (below) (Responses to reviewer’s Comments), (b) an updated manuscript with yellow highlighting indicating changes, and (c) a clean updated manuscript without highlights.

Best regards,

<Chen Li> et al.

Responses to reviewer’s Comments

The authors would like to thank the reviewer for his/her help and efforts by providing constructive comments and valuable suggestions. Changes have been made in the revised manuscript to incorporate most of the reviewer’s suggestions. The changes and justifications for clarifying the reviewer’s comments are summarized as follows:

Reviewer #2: The article is interesting, the work is accurate and clearly described, only a few typing errors and expressions have to be corrected; see below. Some points should be clarified to make the whole work more understandable for the reader.

Concern #1: Introduction reports: “… focussing on bearings of rotating machinery. It is able to distinguish between all eight different”, this expression should be corrected.

Author response: Thank you very much for your constructive comments and valuable suggestions. At the suggestion of reviewers, we corrected the introduction of this paper.

Author action: We adjusted ‘‘ Janssens [19] proposes a novel automatic fault detection system using infrared imaging, focussing on bearings of rotating machinery. It is able to distinguish between all eight different conditions with an accuracy of . ’’ to ‘‘ Janssens [22] proposed a novel automatic fault detection system using infrared imaging, focusing on bearings of rotating machinery. The system is able to distinguish between eight different conditions with an accuracy of . ’’ in introduction, on page 2.

Concern #2: Introduction reports: “… So far, a large number of researches had successfully promoted LC-based passive wireless sensing technology and expanded its applications in many fields …”, this expression should be corrected.

Author response: Thank you very much for your constructive comments and valuable suggestions. At the suggestion of reviewers, we corrected the introduction of this paper.

Author action: We adjusted ‘‘ So far, a large number of researches had successfully promoted LC-based passive wireless sensing technology and expanded its applications in many fields [25-28]. ’’ to ‘‘ Thus far, several studies have successfully promoted wireless LC-based sensing technology and expanded its applications in many fields [33–36]. ’’ in introduction, on page 2.

Concern #3: Experiments 2.2 reports “…on the surface of the inner ring of the tested bearing by screen printing process to form a spiral inductor with an inner diameter of 20 mm and …”, this expression should be corrected.

Author response: Thanks for reviewer’s kind advice. At the suggestion of the reviewer, we keep two decimal places(or two significant figures) for the data in the paper.

Author action: We adjusted ‘‘ Silver paste (9912-K FL) was integrated in situ on the surface of the inner ring of the tested bearing by screen printing process to form a spiral inductor with an inner diameter of  and an outer diameter of , a line width of  and a line pitch of 1mm, with two turns in total. ’’ to ‘‘ Integrate silver paste (9912-K FL) in situ on the surface of the inner ring of the test bearing using the screen printing process to form a spiral inductor with an inner diameter of , outer diameter of , line width of , and a line pitch of 1 mm, with two turns in total. ’’ in experiment 2.2, on page 3.

Concern #4: The temperature unit “℃” in some case is close to the corresponding number, in other cases there is a space in between the number and the T unit, it should be uniform through the text.

Author response: Thank you very much for your constructive comments and valuable suggestions. At the suggestion of the reviewer, we checked the whole paper to make sure there is a space between the number and the unit.

Author action: We adjusted ‘‘ Through the analysis of the mode by Origin software, the diffraction peak can be confirmed at  The purity of silver formation of , ,, and  corresponds to the (), (), (), and () planes of the face centered cubic (FCC) structure. It is also confirmed that at The purity of alumina formed at  , , ,, and  corresponds to the (), (), (), (), () and () planes of face centered cubic (FCC) structure. ’’ to ‘‘ Through the analysis of the mode using the Origin software, the diffraction peak can be confirmed at  The purity of the silver formation at , ,, and  corresponds to the (), (), (), and () planes of the face-centered cubic (FCC) structure, respectively. In addition, it is confirmed that at , the purity of alumina formed at  , , ,, and  corresponds to the (), (), (), (), (), and () planes of the face-centered cubic (FCC) structure, respectively. ’’ in title.

Figure 1. Fabrication of printed temperature sensor using screen printing.

Figure 3. (a) SEM of wireless LC conformal temperature sensor at  for 3 h; (b) SEM of wireless LC conformal temperature sensor at  for 10 h.

Figure 4. (a) XRD of wireless LC conformal temperature sensor at  for 3 h; (b) XRD of wireless LC conformal temperature sensor at  for 10 h.

Figure 6. (a) S11-frequency characteristic curve at different speeds; (b) S11-frequency characteristic curve at different temperatures.

Figure 8. (a) Stability of frequency with temperature atrpm; (b) Stability of S11 with temperature atrpm; (c) Stability of frequency with temperature at  rpm; (d) Stability of S11 with temperature at .

Figure 9. Comparison of nonlinear error () values for five sets of speed measurements at different rotational speeds.

Figure 11. Uncertainty of wireless LC conformal temperature sensor at different rotational speeds.

Concern #5: Result and Discussion reports “… The Ag film is tightly connected with the bearing base, as shown in Figure 2 (c) …”, this expression should be corrected.

Author response: Thank you very much for your constructive comments and valuable suggestions. At the suggestion of the reviewer, we revised the expression.

Author action: We adjusted ‘‘ The Ag film is tightly connected with the bearing base, as shown in Figure 2 (c). ’’ to ‘‘ The Ag film is tightly connected to the bearing base, as shown in Figure 2 (c). ’’ in results and discussion, on page 3.

Concern #6: Conclusions reports “ … Compared with the previous wireless bearing temperature measurement system, the LC conformal passive temperature sensor proposed in this study has higher temperature measurement range and accuracy, and can work under the measured condition for a long time.” This expression should be corrected.

Author response: Thank you very much for your constructive comments and valuable suggestions. At the suggestion of the reviewer, we revised the expression.

Author action: We adjusted ‘‘ Compared with the previous wireless bearing temperature measurement system, the LC conformal passive temperature sensor proposed in this study has higher temperature measurement range and accuracy, and can work under the measured condition for a long time. ’’ to ‘‘ Compared with previous wireless bearing temperature measurement systems, the wireless LC conformal temperature sensor proposed in this study has a higher temperature measurement range and accuracy and can successfully function under the measured condition for an extended period. ’’ in conclusion, on page 10.

Concern #7: The authors affirm: “... the results show that the sensor can work steadily for 10 hours at 300 °C. …” This statement must be supported by the frequency response data, not only on the results obtained with x-rays and SEM.

Author response: Thank you very much for your constructive comments and valuable suggestions. At the suggestion of the reviewer, we added the graph of frequency with time in the high-temperature holding experiment to prove the stability of the designed sensor in long-time high-temperature measurement, as shown in Figure 13.

Author action: We updated the manuscript by adding “ In order to further prove the stability of the designed sensor in long-term high-temperature measurement, we carried out a high-temperature holding experiment at  for 10 hours, and recorded the frequency characteristic curve of the antenna every hour with a network analyzer. As shown in figure 13, the frequency still keeps good stability under long-term high-temperature measurement, while S11 gradually increases with time.

Figure 13. (a) S11-frequency characteristic curve in high-temperature holding experiment at  for 10 hours; (b) Frequency and S11 change with the length of time maintained at .

 ” in the paper.

Concern #8: More information on the Ag film (9912-K FL) should be included in the paper (Materials). The final film seems to be composed by Ag and Al2O3. Author may explain which is the role of Al2O3?

Author response: Thank you very much for your constructive comments and valuable suggestions. At the suggestion of the reviewer, we supplemented the information of the materials. The Ag and Al2O3 shown in Fig. 2 (c) represented the sensor (Ag film made of 9912-K FL) and the bearing (the test bearing material was all ceramic) respectively. Fig. 2 (c) was to show that the sensor can be tightly connected with the bearing, as mentioned in the article, “ the sensor (Ag film) is tightly connected with the bearing base, as shown in Fig. 2 (c) ”. We realized that the instructions in Figure 2 were not clear enough, so we modified Figure 2.

Author action: We updated the manuscript by adding “ The levelling time of ESL 9912-K FL is  at  and the drying time is  at . Furthermore, the firing temperature is  on alumina/beryllia/ceramics. ” in the paper (Materials).

Figure 2. (a) Physical bearing after preparation; (b) Measuring the thickness of Ag film with a step meter; (c) SEM diagram of Ag film and bearing cross section.

Concern #9: Table 1, reporting PARAMETERS OF XRD, is not essential as table, data can be included in the main text.

Author response: Thank you very much for your constructive comments and valuable suggestions. At the suggestion of the reviewer, we added XRD parameters in the main text to replace the original Table 1.

Author action: We updated the manuscript by adding “ In addition, the sensor were characterized using an X-ray diffractometer (Japan Rigaku Miniflex), in which the scanning angle ranged from  to , the scanning speed was , and the test target was copper. ” on page 4, paragraph 1.

Concern #10: Section “Result and discussion” reports the reflection coefficient at the antenna end, parameter, impedance and sensor resonance frequency as 1-4 equations. All the symbols used in the equation must be explained.

Author response: Thank you very much for your constructive comments and valuable suggestions. At the suggestion of the reviewer, we explained all the symbols used in the 1-4 equations. Furthermore, we supplemented the equivalent circuit model of the sensor in Figure 5, so as to understand the meaning of each symbol in equation 1-4.

Author action: We adjusted ‘‘ When the sensor is coupled with the antenna, the reflection coefficient at the antenna end, S11 parameter, impedance and sensor resonance frequency can be expressed as [33,34] ’’ to ‘‘ When the sensor is coupled with the antenna, the reflection coefficient() at the antenna end, S11 parameter, impedance() and sensor resonance frequency() can be expressed as equation 1-4 [43,44]. The circuit equivalent model of the sensor is shown in Figure 5(b), where ,  and  represent the equivalent resistance, equivalent inductance and equivalent capacitance of the antenna equivalent circuit, respectively. ,  and  represent the equivalent resistance, equivalent inductance and equivalent capacitance of LC equivalent circuit, respectively.  represented the mutual inductance coupling coefficient between the two circuits. ’’ in title.

Figure 5(a) Wireless LC passive temperature sensor and its application in wireless measurement system of bearing temperature; (b) Wireless transmission of temperature signal; (c) The coupling between antenna and coil at different angles is simulated by ANSYS.

Concern #11: By keeping the color, of figure 8, matched to the T specification, data comparison is more straightforward and less complicated.

Author response: Thank you very much for your constructive comments and valuable suggestions. At the suggestion of the reviewer, we changed the color in Figure 8 to be consistent with the temperature specification, making the data comparison is more straightforward and less complicated.

Author action:

Figure 8. (a) Stability of frequency with temperature atrpm; (b) Stability of S11 with temperature atrpm; (c) Stability of frequency with temperature at  rpm; (d) Stability of S11 with temperature at .

Concern #12: Authors should clarify in the text how the nonlinear error of the sensor has been evaluated, since the fitting equations reported in figures 7 and 9 are the same, and its meaning.

Author response: There are several differences between Figure 7 and Figure 9. We would like to explain the specific meaning of Figure 7 and Figure 9 to reviewers in detail. Figure 7 (a) is the juxtaposition data fitting of frequency with temperature at different rotating speeds (the frequency data of four groups of rotating speeds are fitted as a straight line with temperature), in order to show that the frequency still has a good linearity with temperature in the rotating environment compared with S11 parameter in Figure 7 (b). As we explained in this article,  It can be seen from the graph that the frequency variation with temperature has good linearity at different rotational speeds, while S11 has poor linearity with temperature.  Figure 9 is the linear fitting of frequency with temperature at different rotational speeds (fitting as 4 straight lines), which is to analyze the influence of rotational speed on frequency with temperature. As we explained in this article,  As can be seen from figure 9, the frequency keeps a good linearity with the change of temperature at different rotational speeds, and the sensitivity of the sensor increases slightly with the increase of rotational speed.

We appreciate for reviewer’s warm work earnestly, and hope that the correction will meet with approval. Thanks for your constructive comment and valuable suggestions once again, and the authors hope to be able to learn more knowledge from you.

Reviewer 3 Report

line 42: "Jhon" ???

line 42: "pm/" meaning?

In general: too many digits indicated for numerical values: use only the significant ones (a few)

Use of "passive": all thermometers are passive device and require some power through them to get a signal (also in your case). You actually mean "wireless": please correct also in the Title.

"10 h duration": why you indicate a duration? Is the lifetime of the sensor limited? Why? Clarify

If the sensor lifetime is limited, is it enough or the monitoring must be always ensured ? In the latter case how can it be ensured?

Author Response

Original Manuscript ID: nanomaterials-1836591

Original Article Title: LC conformal passive temperature sensor based on Ag film (9912-K FL) for bearing temperature measurement

TO: nanomaterials reviewer

RE: Response to reviewers

Dear reviewer,

Thank you for allowing a resubmission of our manuscript, with an opportunity address the reviewers’ comments.

We are uploading (a) our point-by-point response to the comments (below) (Responses to reviewer’s Comments), (b) an updated manuscript with yellow highlighting indicating changes, and (c) a clean updated manuscript without highlights.

Best regards,

<Chen Li> et al.

Responses to reviewer’s Comments

The authors would like to thank the reviewer for his/her help and efforts by providing constructive comments and valuable suggestions. Changes have been made in the revised manuscript to incorporate most of the reviewer’s suggestions. The changes and justifications for clarifying the reviewer’s comments are summarized as follows:

Reviewer #3:

Concern #1: line 42: "Jhon" ???

Author response: Thank you very much for your constructive comments and valuable suggestions. At the suggestion of reviewers, we checked and corrected the introduction of this paper.

Author action: We adjusted ‘‘ Jhon ’’ to ‘‘ Henao-Sepulveda ’’ in line 42.

Concern #2: line 42: "pm/" meaning?

Author response: Thank you very much for your constructive comments and valuable suggestions. We would like to explain the meaning of “pm” in line 42.  pm/℃  refers to the sensitivity of the FBG temperature sensor designed by the authors in the cited reference that the fiber will offset by 35.165 pm for every 1 ℃ change in temperature.

Concern #3: In general: too many digits indicated for numerical values: use only the significant ones (a few)

Author response: Thanks for reviewer’s kind advice. At the suggestion of the reviewer, we keep two decimal places (or two significant figures) for the data in the paper (including the data in the figures).

Author action:

Figure 5(a) Wireless LC passive temperature sensor and its application in wireless measurement system of bearing temperature; (b) Wireless transmission of temperature signal; (c) The coupling between antenna and coil at different angles is simulated by ANSYS.

Figure 7. (a) The frequency at different speeds fits with temperature; (b) The frequency changes with temperature at different speeds.

Figure 9. Comparison of nonlinear error () values for five sets of speed measurements at rotational speed.

Concern #4: Use of "passive": all thermometers are passive device and require some power through them to get a signal (also in your case). You actually mean "wireless": please correct also in the Title.

Author response: Thank you very much for your constructive comments and valuable suggestions. At the suggestion of the reviewer, we revised the title of the paper. Furthermore, we correct all the expressions of wireless LC conformal temperature sensors in this paper.

Author action: We adjusted ‘‘ LC conformal passive temperature sensor ’’ to ‘‘ Wireless LC conformal temperature sensor ’’ in title.

We adjusted ‘‘ In this study, a LC conformal passive temperature wireless measurement sensor was demonstrated for bearing’s temperature measurement. ’’ to ‘‘ In this study, a wireless LC conformal temperature measurement sensor was demonstrated for the temperature measurement of bearings ’’ in introduction.

We adjusted ‘‘ The high-temperature rotating experimental platform also verified the superior performance of the designed LC conformal passive temperature sensor. ’’ to ‘‘ The high temperature rotating experimental platform also verified the superior performance of the designed wireless LC conformal temperature sensor. ’’ in introduction.

We adjusted ‘‘ 2.2 In situ fabrication of conformal passive temperature sensor ’’ to ‘‘ 2.2 In situ fabrication of wireless LC conformal temperature sensor. ’’ in experiment.

We adjusted ‘‘ As shown in Figure , the preparation of LC conformal passive temperature sensor includes the following steps ’’ to ‘‘ As shown in Figure , preparation of the wireless LC conformal temperature sensor includes the following steps ’’ in experiment.

We adjusted ‘‘ Figure 3. (a) SEM of LC conformal passive temperature sensor under  for 3 hours; (b) SEM of LC conformal passive temperature sensor under  for 10 hours. ’’ to ‘‘ Figure 3. (a) SEM of wireless LC conformal temperature sensor at  for 3 h; (b) SEM of wireless LC conformal temperature sensor at  for 10 h. ’’ in experiment.

We adjusted ‘‘ Figure 4. (a) XRD of LC conformal passive temperature sensor under  for 3 hours; (b) XRD of LC conformal passive temperature sensor under  for 10 hours. ’’ to ‘‘ Figure 4. (a) XRD of wireless LC conformal temperature sensor at  for 3 h; (b) XRD of wireless LC conformal temperature sensor at  for 10 h. ’’ in result and discussion.

We adjusted ‘‘ In order to verify the performance of LC conformal passive temperature sensor, we built a platform for wireless measurement of bearing temperature based on LC passive temperature sensor, which mainly includes motor, antenna, tested bearing, support bearing, thermocouple, heating plate, temperature control panel, network analyzer, as shown in Figure 5 (a). ’’ to ‘‘ To verify the performance of wireless the LC conformal temperature sensor, we constructed a platform for the wireless measurement of bearing temperatures based on an LC temperature sensor, which mainly comprises a motor, antenna, test bearing, support bearing, thermocouple, heating plate, temperature control panel, and network analyzer, as shown in Figure 5 (a). ’’ in result and discussion.

We adjusted ‘‘ The simulation results show that the coupling of different angles has almost no influence on the characteristic frequency of the antenna, which indicates that the designed LC conformal passive temperature sensor can realize wireless temperature measurement in rotating environment. ’’ to ‘‘ The simulation results show that coupling at different angles has almost no influence on the characteristic frequency of the antenna, which indicates that the designed wireless LC conformal temperature sensor can realize wireless temperature measurement in a rotating environment. ’’ in result and discussion.

We adjusted ‘‘ By observing and analyzing the graph, it could be seen that the frequency and S11 parameter fluctuate slightly with the rotating speed, which proved that the proposed LC conformal passive temperature sensor is effective in wireless temperature measurement of bearings. ’’ to ‘‘ By observing and analyzing the graph, it can be seen that the frequency and S11 parameter fluctuate slightly with the rotating speed, which proves that the proposed LC wireless conformal temperature sensor is effective in the wireless temperature measurement of bearings. ’’ in result and discussion.

We adjusted ‘‘ and also proves the effectiveness of the proposed LC conformal passive temperature sensor in rotating environment. ’’ to ‘‘ and proves the effectiveness of the proposed wireless LC conformal temperature sensor in a rotating environment. ’’ in result and discussion.

We adjusted ‘‘ which indicates that the proposed LC conformal passive temperature sensor in rotating environment keeps good stability in the whole temperature range. ’’ to ‘‘ which indicates that the proposed wireless LC conformal temperature sensor maintains good stability throughout the temperature range in a rotating environment. ’’ in result and discussion.

We adjusted ‘‘ The proposed LC conformal passive temperature sensor has high measurement accuracy in the whole temperature range and is not affected by rotational speed. ’’ to ‘‘ The proposed LC wireless conformal temperature sensor has high measurement accuracy throughout the temperature range and is not affected by rotational speed. ’’ in result and discussion.

We adjusted ‘‘ The results showed that the proposed LC conformal passive temperature sensor could work stably at  for  hours. The LC temperature sensor was tested at  with a high-temperature rotating experimental platform. ’’ to ‘‘ The results showed that the proposed wireless LC conformal temperature sensor could function stably at  for  h. The LC temperature sensor was tested at  with a high temperature rotating experimental platform. ’’ in conclusion.

We adjusted ‘‘ the LC conformal passive temperature sensor proposed in this study has higher temperature measurement range and accuracy, and can work under the measured condition for a long time. ’’ to ‘‘ the wireless LC conformal temperature sensor proposed in this study has a higher temperature measurement range and accuracy and can successfully function under the measured condition for an extended period. ’’ in conclusion.

Concern #5: "10 h duration": why you indicate a duration? Is the lifetime of the sensor limited? Why? Clarify

Author response: Thank you very much for your constructive comments and valuable suggestions. As we all know, the sensor needs continuous measurement for a long time to verify its performance. In order to verify that the designed sensor can keep long-term stable operation in high temperature environment, we carry out high temperature keeping experiment (working at  for 10 h). We supplemented the graph of the frequency characteristic curve with the high temperature holding time (Figure 13) to prove that the sensor remained stable after a long time of high temperature. Any sensor has a service life, but the service life of the sensor designed in this paper is much longer than 10 h.

Author action: We updated the manuscript by adding “ In order to further prove the stability of the designed sensor in long-term high-temperature measurement, we carried out a high-temperature holding experiment at  for 10 hours, and recorded the frequency characteristic curve of the antenna every hour with a network analyzer. As shown in figure 13, the frequency still keeps good stability under long-term high-temperature measurement, while S11 gradually increases with time.

Figure 13. (a) S11-frequency characteristic curve in high-temperature holding experiment at  for 10 hours; (b) Frequency and S11 change with the length of time maintained at .

 ” in the paper.

Concern #6: If the sensor lifetime is limited, is it enough or the monitoring must be always ensured ? In the latter case how can it be ensured?

Author response: Thank you very much for your constructive comments and valuable suggestions.  There is no doubt that the lifetime of any type of sensor is limited. Ideally, we should give the service life of the sensor. However, due to the limitation of the experimental environment, we can't continuously test the sensor to verify its service life. Therefore, it is necessary to repair or replace the sensor regularly.

We appreciate for reviewer’s warm work earnestly, and hope that the correction will meet with approval. Thanks for your constructive comment and valuable suggestions once again, and the authors hope to be able to learn more knowledge from you.

Round 2

Reviewer 1 Report

No comments.

Reviewer 3 Report

Now all my advices where accepted